# Genome-Wide Identification and Characterization of *G2-Like* Transcription Factor Genes in Moso Bamboo (*Phyllostachys edulis*)

**DOI:** 10.3390/molecules27175491

**Published:** 2022-08-26

**Authors:** Ruihua Wu, Lin Guo, Ruoyu Wang, Qian Zhang, Hongjun Yao

**Affiliations:** 1College of Biological Sciences and Technology, Beijing Forestry University, Beijing 100083, China; 2Guanghua Qidi School, Shanghai 201799, China; 3Institute of Microbiology, Chinese Academy of Sciences, Beijing 100020, China

**Keywords:** moso bamboo, *GLK* genes, phylogenetic relationship, motif analysis, expression profiles

## Abstract

*G2-like* (GLK) transcription factors contribute significantly and extensively in regulating chloroplast growth and development in plants. This study investigated the genome-wide identification, phylogenetic relationships, conserved motifs, promoter cis-elements, MCScanX, divergence times, and expression profile analysis of *PeGLK* genes in moso bamboo (*Phyllostachys edulis*). Overall, 78 putative *PeGLKs* (*PeGLK1*–*PeGLK78*) were identified and divided into 13 distinct subfamilies. Each subfamily contains members displaying similar gene structure and motif composition. By synteny analysis, 42 orthologous pairs and highly conserved microsynteny between regions of *GLK* genes across moso bamboo and maize were found. Furthermore, an analysis of the divergence times indicated that *PeGLK* genes had a duplication event around 15 million years ago (MYA) and a divergence happened around 38 MYA between *PeGLK* and *ZmGLK*. Tissue-specific expression analysis showed that *PeGLK* genes presented distinct expression profiles in various tissues, and many members were highly expressed in leaves. Additionally, several *PeGLKs* were significantly up-regulated under cold stress, osmotic stress, and MeJA and GA treatment, implying that they have a likelihood of affecting abiotic stress and phytohormone responses in plants. The results of this study provide a comprehensive understanding of the moso bamboo GLK gene family, as well as elucidating the potential functional characterization of *PeGLK* genes.

## 1. Introduction

Chloroplasts contain the green pigment chlorophyll, which can carry out photosynthesis and is thus essential in plant life processes [1,2,3]. Recent research has supported the idea that chloroplasts are derived from protoendosymbiosis events associated with these cyanobacteria [4,5,6]. Furthermore, photosynthetic organs are assembled under the coordinated regulation of chloroplasts and nucleus. Plastids function not only in photosynthesis and nutrient storage, but also in the compartmentalization of various metabolic intermediates [7]; for example, most amino acids, all fatty acids, purines and pyrimidine bases, terpenes and various pigments, and hormones are synthesized in plastids. In addition, proplastids in plant meristem cells (leaf sheaths in dark cotyledons) are converted to chloroplasts of mesophyll cells under light, while *GLK* genes participate in the regulation and control of chloroplast formation in transition and early maturity and are indispensable in the development of angiosperm chloroplasts [2,7,8,9,10].

*GLK* genes encode members of the GARP superfamily of nuclear transcription factors [11] defined by Golden2 in maize (*Zea mays* L.), Response regulator-B (ARR-B) proteins in *Arabidopsis thaliana* [12], and phosphate starvation response1 (PSR1) protein in Chlamydomonas. Most GLK proteins contain two conserved domains: an Myb-DNA binding domain (DBD, containing an HLH motif) and a C-terminal domain (containing a conserved GCT box) [13,14].

*GLK* genes promote the production of plant chloroplasts, and optimise photosynthesis under different biotic and abiotic environmental stress conditions [15,16]. For instance, *AtGLK1* and *AtGLK2* function in the formation and development of chloroplasts redundantly in *Arabidopsis* [17,18], and *AtGLK1 OE* increases resistance to *Fusarium graminearum*, a wide range of host pathogens that cause significant losses in cereal crops. In contrast, the *glk1 glk2* double mutant provides resistance to *Hyaloperonospora arabidopsidis* (Hpa), implying a potential role of *GLKs* in plant defense and disease resistance [19]. In addition, *GLK1* and *GLK2* participate in chloroplast growth and development of C3 photosynthetic plants redundantly [15]. In tomato (*Solanum lycopersicum*), *SlGLK2 OE* enhances fruit photosynthesis and chloroplast development gene expression, resulting in elevated carbohydrates and carotenoids in ripe fruit, which can lead to strong resistance to cold, drought, heat and other abiotic stresses [20]. In maize, the spatial compartmentalization of *ZmGLK1* transcripts and G2 might function as a specialization that was essential to the chloroplast development of mesophyll cells and distinct bundle sheath in plant tissues [14,17].

Moso bamboo (*Phyllostachys edulis*) is a rapidly growing forest product that is widely used to manufacture paper, art ware, and food resources, with multiple economic, ecological, and cultural values in China and many other countries, sees a possibility for gene identification and function analysis that will help with it contributing on an even wider spectrum with the completion of the genome sequence [6,20,21]. The *GLK* gene family, previously identified and described within certain plants, for example, maize [22], *Arabidopsis* [6], tomato [23], and tobacco [24]. However, there are scarcely any reports on the characteristics and functions of *GLKs* in *Phyllostachys edulis*. Thus, this study identified 78 putative *PeGLK* genes and conducted a systematic analysis, including phylogenetic relationships, gene and protein structures, conserved motifs, promoter *cis*-elements, chromosomal localization, MCScanX, evolutionary patterns, and tissue-specific expression. The expression patterns of *PeGLKs* in response to abiotic stress (cold and osmotic) and phytohormone (MeJA and GA) treatment were analyzed as well. These results are valuable insightful for further understanding of potential roles of the moso bamboo *GLK* gene family.

## 2. Materials and Methods

### 2.1. Plant Material and Growth Conditions

For this experiment, moso bamboo (*Phyllostachys edulis*) seeds, which were provided by Gongcheng Yao Autonomous County, Guangxi Zhuang Autonomous Region, China, were germinated and grown under daylight conditions of 16 h light/8 h dark and maintained at 28 °C and 80% relative humidity in an artificial growth chamber for 100-day-old. The 100-day-old plants were subjected to cold stress, osmotic stress, and MeJA and GA treatments. Specifically, *Phyllostachys edulis* plants were placed in a 4 °C growth chamber and leaves were gathered under cold stress at 0, 1, 3, 6, 12, and 24 h, and under osmotic stress at 0, 1, 2, 3, 6, and 12 h after treatment with pure water containing 20% polyethylene glycol (PEG) 6000. For hormone treatments, a solution of 100 µM Methyl jasmonic (MeJA) and 300 mg/L gibberellic acid (GA) was sprayed onto seedlings based on the requirements, and seedlings were sampled randomly at 0, 1, 3, 6, and 12 h. Negative control plants were treated in a 28 °C artificial growth chamber and sprayed with distilled water.

### 2.2. Identification of PeGLK Genes

*Phyllostachys edulis* protein and nucleic acid sequences were acquired from the moso bamboo genome database (http://parrot.genomics.cn, accessed on 10 March 2022). Previously reported GLK protein sequences of *Arabidopsis* were used to inquire the *Phyllostachys edulis* protein database with BLASTP tools on 18 March 2022 (http://www.ncbi.nlm.nih.gov/tools/primer-blast/, accessed on 18 March 2022). Then, the redundant sequences were deleted according to the BLAST results of a ClustalW [25] alignment, and Pfam (http://pfam.xfam.org/, accessed on 25 March 2022) was used to identify putative PeGLK members in moso bamboo *(Phyllostachys edulis*) by analyzing conserved domains of these proteins. Detailed information of the *PeGLK* genes, including gene IDs, physical positions, gene and protein sequences, and coding sequences (CDSs), were retrieved from the bamboo database. The ExPASy online tool (http://www.expasy.ch/tools/pi_tool.html, accessed on 10 May 2022) was used to determine the physical parameters of *PeGLK* genes, including molecular weight (MW), isoelectric point (pI), and open reading frame length. 

### 2.3. PeGLK Sequence Alignment and Phylogenetic Analysis

To identify the domain organization of 78 PeGLK proteins, multiple alignments of PeGLK domain-containing sequences were performed using ClustalW [25], and a phylogenetic tree based on the complete PeGLK sequences was constructed using the neighbor-joining method by MEGA 7.0 with 1000 bootstrap replications (https://megasoftware.net/, accessed on 10 April 2022). In addition, the combined phylogenetic tree of PeGLK and ZmGLK protein sequences was also generated with the N-J method.

### 2.4. Gene and Protein Structure Analysis

The exon–intron structures were obtained by contrasting the CDSs and relevant genomic DNA sequences of *PeGLK* genes from GSDS (http://gsds.cbi.pku.edu.cn/, accessed on 14 April 2022). Conserved motifs present in the PeGLK proteins were examined by the online MEME tool (http://meme-suite.org/tools/meme, accessed on 18 April 2022) [26]. The optimum width of motifs was limited to 5–50, and the maximal number of motifs was set to 10 residues. Then, we used Pfam tools to determine the motif annotations. The protein homology models were predicted by aligning GLK protein sequences with the HMM-HMM search in intensive mode on the Phyre2 website (http://www.sbg.bio.ic.ac.uk/phyre2/html/page.cgi?id=index, accessed on 26 April 2022) [27]. 

### 2.5. Chromosomal Location and Synteny Analysis

According to the position information in the genome annotation file (GFF file) of the moso bamboo (*Phyllostachys edulis*) genome database, 78 *PeGLK* genes were located on 23 moso bamboo chromosomes. MapInspect software (https://mybiosoftware.com/mapinspect-compare-display-linkage-maps.html, accessed on 3 May 2022) was then used to graphically depict these *PeGLK* genes. Possible segmental and tandem duplication events were investigated, and potential duplication events were identified by MCScanX software (http://chibba.pgml.uga.edu/mcscan2/#tm, accessed on 10 May 2022) [28]. 

To perform a collinearity analysis of PeGLK protein in *Phyllostachys edulis*, BLAST was used to compare the entire sequence, with a cutoff of truncated E-value of 1 × 10^−20^ [29]. The collinearity analysis of moso bamboo and maize also used the same truncated E-value. MCScanX software was used to produce collinearity blocks for the entire genome by calculating the BLASTP results [28]. 

### 2.6. Calculation of Ka/Ks Values

Ka and Ks values represent the numbers of nonsynonymous substitutions per nonsynonymous site and synonymous substitutions per synonymous site [30]. The KaKs Calculator 2.0 (https://ngdc.cncb.ac.cn/biocode/tools/BT000001, accessed on 25 May 2022) was used to calculate Ka and Ks values with the NY model, and the ratios were then calculated using DnaSP5 [31,32]. Additionally, the divergence time (T) was calculated using the formula T = Ks/2 λ (λ = 6.5 × 10^−9^) by transforming the duplication event date for each gene pair [33].

### 2.7. Putative Promoter Region Analysis of PeGLK Genes

The 2000 bp upstream sequences of *PeGLK* genes were selected to determine the *cis*-elements in the putative promoter regions. PlantCARE (http://www.dna.affrc.go.jp/PLACE/, accessed on 1 June 2022) was used to predict the putative *cis*-elements in the gene promoters [34,35]. Then, the *cis*-elements that responded to abiotic stress and phytohormone treatments were screened out.

### 2.8. PeGLK Expression Profiles, RNA Extraction, and qRT-PCR Analysis

According to the plant’s growth characteristics, four different tissues (leaf, root, stem, and rhizome) of a 100-day-old moso bamboo were used as samples. Then, the tissue-specific expression profile for each *PeGLK* gene was determined, taking the tonoplast intrinsic protein 41 (*TIP41*) gene as the internal reference standard, and the relative expression level of each gene was calculated via the 2^−ΔΔCt^ method [36,37]. The heatmap of *PeGLK* gene expression was drawn using the Amazing Heatmap module in TBtools (https://github.com/CJ-Chen/TBtools/releases, accessed on 15 June 2022) for the four *Phyllostachys edulis* tissues [38].

According to the similarity with the reference gene in the phylogenetic tree, 36 genes were used for qRT-PCR. The 36 primers of *PeGLK* genes were designed by the NCBI Primer-BLAST tool (http://www.ncbi.nlm.nih.gov/tools/primer-blast/, accessed on 25 June 2022) to amplify 150–300 bp PCR products (Appendix A). Total RNA was collected from samples of four different tissues (leaf, stem, rhizome, and root) using an RNA Easy Fast Plant Tissue Kit (Tiangen, Beijing, China) based on the manufacturer’s instructions. First-strand cDNA was synthesized using total RNA by an EasyPure Genomic DNA Kit (TransGen Biotech, Beijing, China) [39]. Reaction was performed by the Light Cycler 480 SYBR Green Master Mix (TaKaRa, Dalian, China) on a Bio-Rad CFX96, and the program was used as follows: 95 °C for 60 s, followed by 40 cycles of 95 °C for 10 s, and 60 °C for 30 s. 

## 3. Results

### 3.1. Identification of PeGLK Genes in Phyllostachys edulis

To identify the moso bamboo *GLK* gene family members, previously reported Arabidopsis AtGLK1 (AT2G20570) and AtGLK2 (AT5G44190) protein sequences [17] were used as BLASTP queries to search for available protein databases in the moso bamboo genome bank. We identified 78 *PeGLK* candidate genes, most of which were used to confirm the presence of the Myb DNA-binding domain (DBD) and C-terminal domain (containing a conserved GCT box) using the Pfam database. Except for the conserved GLK Myb DNA-binding domain, the members of each subfamily have suspected functional diversity, represented by unique motifs. For instance, among groups 1–7, each has an Myb-CC-LHEQLE domain, which is a type of Myb-like domain. Group 10 has an REC-typeB-ARR-like domain (Figure 1). To discern the similarities between moso bamboo GLK domains, the domain sequences of 78 PeGLK proteins were blasted with the DNAMAN 8 platform (Appendix A). The results showed that PeGLKs were conserved in two regions of a putative Myb DNA-binding domain, with the HLH structure of the first helix containing the initial sequence, PELHRR, and the second helix containing NI/VASHLQ, which was consistent with the GLK members in *Arabidopsis*, maize [22], tobacco, and tomato [40].

Basic information about *PeGLK* genes, such as the accession number, position, and physicochemical parameters, is presented in Table 1. The molecular weight (MW) varies from 18.08 to 75.00 kDa and the length of CDSs ranged from 486 to 2079 bp, and encoded sequences ranged from 161 to 692 aa. In addition, the theoretical isoelectric point (pI) ranged from 5.51 to 10.43. 

### 3.2. Phylogenetic and Exon-Intron Structure Analysis of GLK Genes 

From a phylogenetic tree formulated on the aligning of 137 sequences of GLK proteins from moso bamboo (78) and maize (59), the phylogenetic relationships of GLK proteins among different species were derived. The characteristics of *ZmGLK* genes from maize are listed in detail in Appendix A. In this phylogenetic tree, *GLK* family members were classified into 16 groups based on evolutionary relationships and motif analysis, and poplar *GLK* family members were allocated into 13 groups (G1–G13, but not G14–G16). The numbers of *PeGLKs* in different groups were uneven: groups 1 to 13 contained 9, 4, 7, 5, 3, 4, 2, 15, 14, 7, 3, 6, and 2 genes, respectively (Figure 2). 

To further explore the evolutionary relationships of *PeGLK* genes, a single phylogenetic tree was constructed using the moso bamboo PeGLK protein sequences. PeGLK proteins were also classified into 13 distinct subfamilies, in accordance with the phylogenetic tree constructed by moso bamboo and maize. The exon–intron predictions were identified by the online Gene Structure Display Server (GSDS) tool. As shown in Figure 3, the number of exons in different subfamilies varied from 1 to 12, and most *PeGLK* genes (91%) had five or more exons. In addition, most genomic sequences contained upstream and downstream sequences, except for *PeGLK5*, *PeGLK13*, *PeGLK48*, *PeGLK50*, *PeGLK67*, *PeGLK68*, *PeGLK69*, and *PeGLK77*, whereas *PeGLK12* and *PeGLK41* had only upstream sequences, and *PeGLK19*, *PeGLK29*, *PeGLK36*, *PeGLK51*, *PeGLK70*, and *PeGLK74* had only downstream sequences. In general, the vast majority of putative paralogous had significant changes in structural organization and intron/exon numbers, except for three paralogous (*PeGLK5*/*PeGLK6*, *PeGLK62*/*PeGLK65*, *PeGLK48*/*PeGLK67*), which had the same intron/exon numbers but different intron lengths.

### 3.3. Identification of Conserved Sequence Motifs and Homology Modeling in PeGLK Genes 

To evaluate the phylogenetic kinship, we examined the conserved motifs of 78 PeGLK proteins using the MEME software (Figure 4). In Appendix A the eight distinct motifs found are shown, with their specific characteristics noted. Functional annotation was made for six putative motifs with the Conserved Domain Database (CDD), in which motifs 1 and 2 were defined as Myb-SHAQKYF, motif 3 was Myb-CC-LHEQLE, and motifs 5, 6, and 8 were REC-type B-ARR-like. However, the remaining two putative motifs did not have functional annotations. Protein family members within the same group shared similar or identical components and spatial distributions, which implies that these proteins have similar functions. For instance, all of the PeGLK proteins were characterized by motif 1 in the Myb DNA-binding domain, with an HLH structure (the first helix contains initial sequence PELHRR and invariably comprises 14 amino acids, and the second helix contains an initial NI/VASHLQ motif. These helices are separated by a 22 amino acid loop.). Additionally, there was specificity within different subfamilies. For example, motif 5 was only present in subfamilies 1, 2, 3, 4, 5, 6, and 7, and motifs 5, 6, 8 only appeared in subfamily 10. 

The structure and homology modeling of PeGLK proteins were determined by Phyre2, and alignment of the protein sequences with HMM-HMM search was conducted in intensive mode [41]. As a result, all 78 PeGLK proteins could be modeled with certainty. As shown in Figure 5, 10 PeGLKs (PeGLK5, PeGLK12, PeGLK13, PeGLK29, PeGLK49, PeGLK51, PeGLK53, PeGLK70, PeGLK71, and PeGLK74) had 100% of their predicted length modeled with >98% confidence.

### 3.4. Physical Locations and Duplication Events of Pelf Genes in Phyllostachys edulis

The 78 *PeGLKs* were unevenly distributed on 23 moso bamboo chromosomes (Chr2-Chr24), except Chr1 (Figure 6). Chromosomes 6 and 8 contained eight *PeGLK* genes, the highest number; chromosomes 15, 17, and 21 had seven, and 20 had five, whereas there was only one *PeGLK* gene on chromosomes 2, 5, 12, 14, and 18. Interestingly, Chr13 was the longest chromosome with only four *PeGLK* genes. These results indicate that the chromosome length was not proportional to the number of genes. To explore the potential mechanism of the *PeGLK* gene family, the MCScanX program was used to investigate potential segmental and tandem duplication events. As shown in Figure 7A, 31 segmental duplicated pairs and one tandem duplicated pair of *PeGLK* genes were identified in a synteny map (Table 2). The 31 segmental duplicated pairs presented a biased distribution pattern, and no pairs were distributed on chromosome 1. These results suggest that moso bamboo *PeGLKs* may have arisen mainly from segmental duplication events.

To get additional insight into PeGLK’s evolutionary orthologous relationships, a synteny map was plotted between moso bamboo and maize. As shown in Figure 7B, 42 orthologous pairs (Pe-Zm) of moso bamboo and maize were determined (Table 3). Across the two, highly conserved microsynteny was observed in the regions of *PeGLK* genes, especially in Pe3 and Zm5, Pe7 and Zm6, Pe15 and Zm1, Pe17 and Zm5, and Pe21 and Zm1, all with three synteny genes.

### 3.5. Evolutionary Patterns and Divergence Times of GLK Genes in Moso Bamboo and Maize

To explore the selective constrains for the duplicated *PeGLK* gene pairs, a comprehensive analysis of the Ka/Ks ratio and Ks values was conducted using full-length sequences to estimate divergence times. As shown in Figure 8 and Table 2, the distribution of calculated Ks values of paralogous pairs (Pe-Pe) showed an average value of ~0.2, indicating that *PeGLK* genes underwent a large-scale duplication event circa 15 MYA. A previous study estimated that the timing for whole-genome duplication of moso bamboo was 7-12 MYA [42], implying that large-scale duplication of *PeGLK* genes occurred earlier [33]. In addition, for the orthologous maize–moso bamboo pairs, the averaged Ks value was ~0.5 (Figure 8 and Table 3), indicating *GLK* genes differentiated 38 MYA. A comparison with a previous study revealed 42–52 divergence times between moso bamboo and maize, implying that *GLK* genes encountered gene evolution before the isolation of maize. In principle, Ka/Ks ratios greater than, equal to, and less than 1 indicate accelerated evolution with positive, neutral, and negative or stabilizing selection, respectively [26,43]. The Ka/Ks ratios of the Pe-Pe (Table 2) and Pe-Zm (Table 3) genomes were less than 1, which indicates that *GLK* genes experienced highly positive purifying selection among moso bamboo–maize and moso bamboo. 

### 3.6. PeGLK Expression Levels in Different Tissues

Tissue-specific gene expression profiles provide critical clues for exploring gene function. In order to characterize the expression levels of *PeGLKs*, we analyzed the transcription levels of 36 *PeGLK* genes selected as representatives of each subfamily in four different tissues (leaf, stem, rhizome, and root) of moso bamboo (Appendix A). As shown in Figure 9, the vast majority of *PeGLK* genes (except for *PeGLK6*, *PeGLK7*, *PeGLK52*, *PeGLK61*, *PeGLK63*, *PeGLK68*, *PeGLK71*) in the leaf, nine genes (*PeGLK47*, *PeGLK51*, *PeGLK52*, *PeGLK60*, *PeGLK63*, *PeGLK65*, *PeGLK66*, *PeGLK69*, *PeGLK71*) in stem, four genes (*PeGLK44*, *PeGLK52*, *PeGLK53*, *PeGLK68*) in rhizome, and 12 genes (*PeGLK6*, *PeGLK7*, *PeGLK20*, *PeGLK21*, *PeGLK28*, *PeGLK36*, *PeGLK37*, *PeGLK39*, *PeGLK53*, *PeGLK61*, *PeGLK72*, *PeGLK78*) in root presented high expression levels. Interestingly, the expression level of most *PeGLKs* was significantly higher in leaf than in the other tissues. We also defined that some genes showed a tissue-specific expression pattern. For example, *PeGLK20*, *PeGLK21*, *PeGLK28*, *PeGLK36*, *PeGLK37*, *PeGLK39*, *PeGLK72*, and *PeGLK78* displayed high expression levels in leaves and roots, but low levels in stems and rhizomes. *PeGLK52* exhibited high expression levels in stems and rhizomes, but low levels in leaves and roots. *PeGLK47*, *PeGLK51*, *PeGLK60*, *PeGLK65*, *PeGLK66*, and *PeGLK67* presented high expression levels in leaves and stems, but low levels in rhizomes and roots. *PeGLK44* and *PeGLK70* displayed high expression levels in leaves and rhizomes, but low levels in stems and roots. In addition, only *PeGLK63* and *PeGLK67* were prominently expressed in stems, and only *PeGLK68* was prominently expressed in rhizomes, whereas *PeGLK6*, *PeGLK7*, *PeGLK21*, *PeGLK36*, *PeGLK61*, and *PeGLK78* were relatively higher in roots, *PeGLK63* and *PeGLK71* in stems, and *PeGLK52* and *PeGLK68* in rhizomes than in the other tissues. Among our previously identified paralogous genes, five pairs (*PeGLK60* and *PeGLK63*, *PeGLK62* and *PeGLK65*, *PeGLK67* and *PeGLK68*, *PeGLK70* and *PeGLK72*) shared the same or similar expression patterns in the four tissues, revealing the same evolutionary fate of duplicated genes. Above all, *PeGLKs* displayed diverse expression profiles in different tissues, which provides insight into the role of *PeGLKs* in multiple growth and development of *Phyllostachys edulis*.

### 3.7. Expression Profiles of GLK Genes under Abiotic Stress and Phytohormone Treatment

To determine the *cis*-regulatory elements in the putative promoter regions, we used PlantCARE to detect the 2000 bp upstream sequences of *PeGLK* genes [44]. Results portray that many *cis*-regulatory elements corresponding to LTRE (cold-responsive element), DRE (drought-responsive element), MeJA, and GA were detected, indicating that *PeGLKs* participate in plant development and stress responses (Figure 10). 

Several *GLK* genes have been reported to respond to cold and osmotic stresses in maize [22], tobacco [45], and tomato [24]. To investigate whether the expression of *PeGLK* genes was affected by abiotic stress and phytohormone treatment, we detected the dynamic expression levels of 13 genes (*PeGLK1*, *PeGLK7*, *PeGLK21*, *PeGLK36*, *PeGLK40*, *PeGLK48*, *PeGLK50*, *PeGLK53*, *PeGLK60*, *PeGLK61*, *PeGLK67*, *PeGLK70*, and *PeGLK72*) as representatives of each subfamily under cold and osmotic stress and MeJA and GA treatment (Figure 11 and Figure 12, and Appendix A). 

In the cold stress treatment, eight genes (*PeGLK1*, *PeGLK21*, *PeGLK36*, *PeGLK40*, *PeGLK53*, *PeGLK60*, *PeGLK67*, and *PeGLK72*) were obviously up-regulated compared with 0 h. For example, the expression of two genes (*PeGLK40* and *PeGLK60*) peaked at 12 h, and *PeGLK60* was the most highly expressed (>80-fold) after 12 h of treatment. Four genes (*PeGLK7*, *PeGLK36*, *PeGLK53*, and *PeGLK72*) gradually increased over time and peaked at 24 h. Seven genes (*PeGLK1*, *PeGLK21*, *PeGLK48*, *PeGLK50*, *PeGLK61*, *PeGLK67*, and *PeGLK70*) showed slight (<5-fold) changes in response to cold stress treatment. 

In the osmotic stress treatment, five genes (*PeGLK7*, *PeGLK36*, *PeGLK53*, *PeGLK67*, and *PeGLK70*) were obviously up-regulated compared with 0 h. For instance, two genes (*PeGLK53* and *PeGLK70*) gradually accumulated and peaked at 6 h, especially *PeGLK70*, which showed an expression level more than 50-fold higher at 6 h, whereas the expression levels of *PeGLK7*, *PeGLK36*, and *PeGLK67* presented parabolic trends and were highest at 12 h. 

In the MeJA treatment, four genes (*PeGLK1*, *PeGLK36*, *PeGLK53*, and *PeGLK70*) showed the highest expression at 12 h, and then a gradually increasing trend. The expression levels of four genes (*PeGLK7*, *PeGLK40*, *PeGLK50*, and *PeGLK61*) were highest at 6 h, especially *PeGLK7*, which presented more than 60-fold higher expression at 6 h. In the GA treatment, two genes (*PeGLK50* and *PeGLK53*) were up-regulated, and *PeGLK53* exhibited an expression level more than 35-fold higher at 12 h.

In total, 46.2, 38.5, 61.5, and 15.4% of genes, including six cold stress-related genes, five osmotic stress-related genes, eight MeJA-related genes, and two GA-related genes, were differently expressed; the detailed gene lists are shown in Appendix A. However, only the expression of *PeGLK53* showed significant changes in response to the two abiotic stresses (Figure 11) and two phytohormone treatments (Figure 12). Additionally, the above results coincide with the promoter analysis of *PeGLK* members, indicating the widespread presence of several osmotic, cold, MeJA, and GA responsive *cis*-elements. Overall, these results imply that some *PeGLK* genes have potential functions in response to abiotic stress and phytohormone treatments.

## 4. Discussion

The function of *GLK* genes was first analyzed in maize and *GLK* genes were served as transcription factors. The *GLK* gene sequences were only found in photosynthetic eukaryotes such as green algae and higher plants, not present in the genome of cyanobacteria, which indicated that *GLK* genes were involved in the formation and development of chloroplast [45]. Members of the *GLK* gene family were characteristic groups in nuclear coding genes, and general feature of organisms with *GLK* genes were associated with chloroplasts and chlorophyll. Chloroplasts convert light energy into chemical energy and play a significant role in plant growth and development; thus, the study of *GLK* genes has become an important prospect of investigation [46].

### 4.1. GLKs in Phyllostachys edulis

The *GLK* transcription factor, which is a member of the newly classified GARP superfamily [14], plays an essential role in the formation and development of chloroplasts [11,47]. As reported in previous studies, detailed features and functions of *GLK* genes have been uncovered in *Arabidopsis* [6], maize [22], tobacco [24], and tomato [23]. However, little was known about the members of the moso bamboo GLK family until now. Here, 78 putative *PeGLK* genes were identified, which is 22 more than maize (59), and there are 30 and 42 times as many genes as tomato (54) and sorghum (45), respectively. The higher number of *PeGLK* genes was in accord with the genome duplication event that occurred in *Phyllostachys edulis* [33,48]. According to the phylogenetic analysis of grouping with maize sequences, the predicted moso bamboo *GLK* family members were divided into 13 groups, G1-G13. However, there were no orthologous genes in maize G14 and G15, which suggests divergence between maize and moso bamboo. In addition, the *PeGLKs* belonging to the same subfamilies displayed high similarity based on their domain structures and number of exons/introns, and these identical results indicate that the *PeGLK* groupings are relatively reliable.

### 4.2. Physical Locations and Divergence of GLKs in Moso Bamboo and Maize

According to the chromosome location analysis, *PeGLKs* were extensively and unevenly spread across 31 chromosomes of *Phyllostachys edulis*, which may be because of insertions, deletions, duplications, and reversions. Previous reports have shown that gene duplication is an important means of gene expansion in the process of plant genome evolution and is also essential for organisms to adapt to various environments during development and growth [49,50]. Thus, we performed genomic collinearity analysis, and found that the moso bamboo genome has 32 pairs of duplicated *PeGLK* genes, including 31 segmental and one tandem duplicated pair. Segmental duplication was obviously the major method of expansion of the moso bamboo *GLK* gene family. Segmental duplication has been shown to be more universal than tandem replication and to play a significant role in long-term evolution [22,51,52,53]. Otherwise, syntheny analysis of the moso bamboo genome with maize sequenced plant genomes indicated significant collinearity of family members between bamboo and monocot maize, which is in accord with the evolutionary relationship between dicotyledons and monocotyledons.

To get more insight into the *Phyllostachys edulis* macroevolutionary profile and divergence times, Ks and Ka for paralogous and orthologous gene pairs were estimated and Ks and Ka/Ks values were calculated for each gene pair. The Ks values showed that *Phyllostachys edulis* had a large-scale duplication event about 15 MYA, and the divergence time for orthologous gene pairs (Pe-Zm) was approximate 38 MYA. It has also been shown that a whole genome duplication event occurred in *Phyllostachys edulis* 7-12 MYA and the divergence time between moso bamboo and maize was 35-42 MYA [33]. Compared with the previous reports, our results suggest that the moso bamboo *GLK* gene family underwent an earlier large-scale duplication event and diversified earlier than maize. In general, the Ka/Ks ratio was available to determine the selective pressure of choice acts on the coding sequences [37]. In the current study, the Ka/Ks ratios for Pe-Pe and Pe-Zm gene pairs were both <1, which imply that the homologous *PeGLK* gene pairs experienced strong purifying selection during the evolutionary process [26,43].

### 4.3. Potential Functions of PeGLKs in Moso Bamboo Development and Stress Responses

*GLK* genes in maize play different roles in the growth and development of chloroplasts in mesophyll cells and bundle sheaths, and they also guide chloroplast development into monomorphic form in *Arabidopsis* [14,17]. Here, we examined the expression levels of 36 putative *PeGLK* genes in four different tissues of *Phyllostachys edulis* and found that great majority genes exhibited high expression levels in the leaf. In view of the important function of chloroplasts in the conversion of light energy into chemical energy in plant growth and development [54], these results imply that *PeGLK* might be involved in the development of chloroplasts, and the high expression level in leaves might satisfy the active metabolism or gene activity. Moreover, *ZmGLK47* was reported to have high expression in all maize tissues, related to the formation and development of chloroplasts [22]. *PeGLK67*, as an ortholog to *ZmGLK47* in *Phyllostachys edulis*, exhibits the same protein structure and expression patterns. 

Plant genomes include various stress-related genes which enable them to respond to diverse environmental conditions [54]. Additionally, the *cis*-elements of promoter regions largely determine the stress-response gene expression profiles that help plants adapt to adverse conditions, and are also closely linked with multiple stimulus-response genes [55,56]. In view of the function of *cis*-elements in abiotic stress resistance, we examined the *cis*-elements of *PeGLKs* and discovered a high number of DRE, LTRE, MeJA, and GA responsive sequences in *PeGLK* promoters, which indicates the significant role of *PeGLKs* in stress responses [22]. Plants produce many signaling molecules in response to abiotic stresses and phytohormone, treatments such as low temperature, drought, salinity, ABA, MeJA, GA, and SA. To date, an association of plant *GLK* with abiotic and phytohormone stress responses has been discovered in many species [6,22,23,24]. Thus, we examined the expression of 13 selected *PeGLK* genes in response to cold, osmotic, MeJA, and GA stress in *Phyllostachys edulis*. Previous studies have shown that orthologous genes in different species were conservative in their gene functions, and paralogous genes present different functions owing to gene duplication [57]. Compared with previous studies, we found that the expression of *PeGLK36* and its ortholog in maize, *ZmGLK1*, exhibited similar patterns in their cold and osmotic stress responses [22]. Conversely, the expression of *PeGLK67* and its ortholog in maize, *ZmGLK50*, showed opposite patterns, indicating that *PeGLKs* may have lost or acquired functions in the process of evolution. Additionally, previous reports have also shown that *AtGLK1* may be involved in the SA and JA signaling pathway in disease prevention [19], and *OsGLK1* is related to resistance to pathogen invasion [58]. Further research on *PeGLKs* might provide more insight into stress-resistant and disease-resistant *Phyllostachys edulis* varieties.

## 5. Conclusions

In this study, 78 moso bamboo *GLK* family genes were identified and could be divided into 11 subfamilies. Members of the same subfamily displayed similar gene structures and motif compositions, implying that *PeGLK* genes were conserved in the process of evolution. Furthermore, 78 *PeGLK* genes were unevenly distributed across the 23 moso bamboo chromosomes. Evolutionary analysis showed that segmental duplication was the main reason for the expansion of moso bamboo *GLK* gene family. The expression profiles of moso bamboo *GLK* genes indicated that *PeGLKs* play significant roles in different tissues of moso bamboo growth and development, and *PeGLKs* was highly expressed in leaf than in other tissues. The expression patterns of *PeGLK* genes under cold stress, osmotic stress, and MeJA and GA treatments offer a different viewpoint about the function of moso bamboo *GLKs* in responses to different abiotic stress and phytohormones treatments. Taken together, these results may be helpful to select suitable candidate genes for further cloning and provide valuable information for functional analysis of *PeGLK* genes.

## Figures and Tables

**Figure 1 molecules-27-05491-f001:**
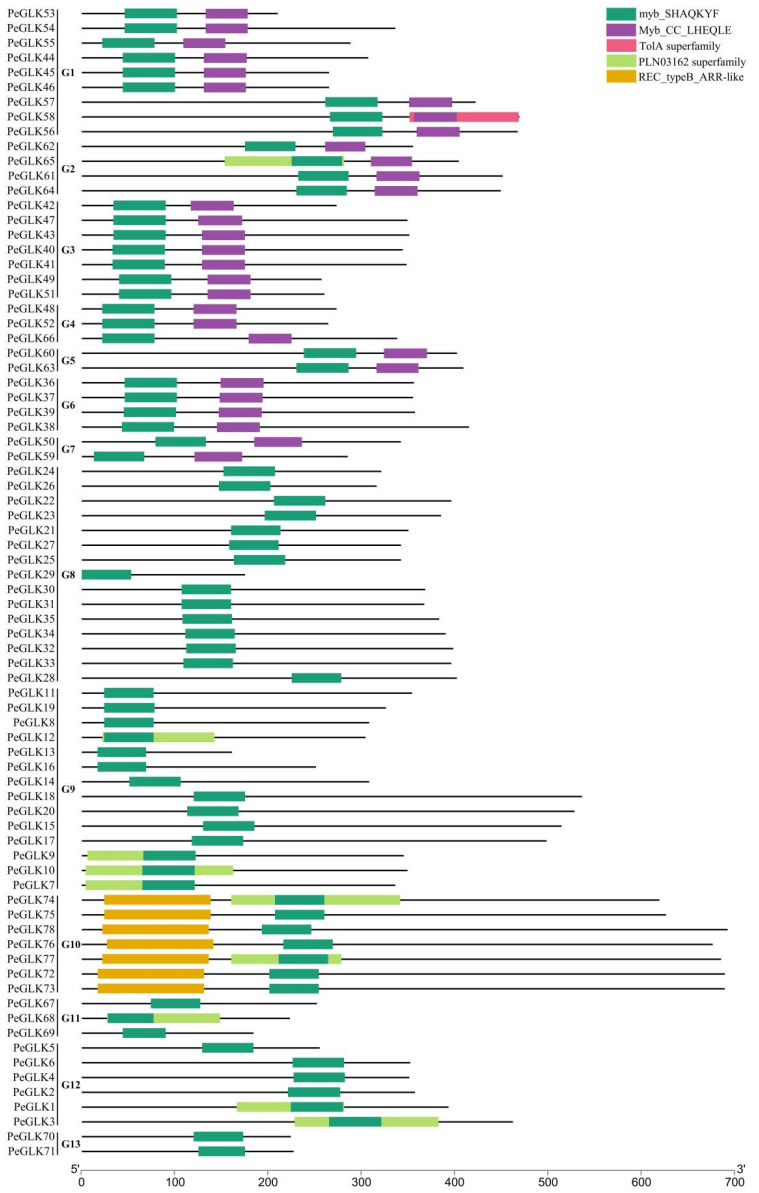
Division of 78 *PeGLK* genes into 13 groups (G1–G13) based on predicted domain structures. Domains are represented by different colored boxes.

**Figure 2 molecules-27-05491-f002:**
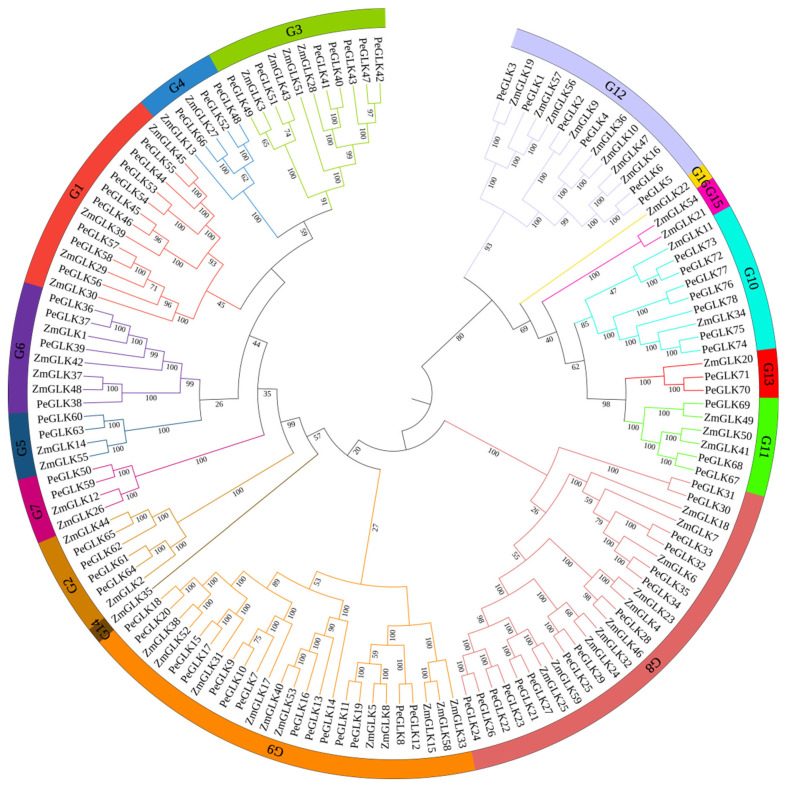
Phylogenetic analysis of GLK proteins of moso bamboo and maize. Tree was constructed using neighbor-joining method with MEGA7.0. Each color represents one group in the branches, and 16 groups were found in total. The numbers at nodes indicate the bootstrap values per 1000 replicates determined by the neighbor-joining method.

**Figure 3 molecules-27-05491-f003:**
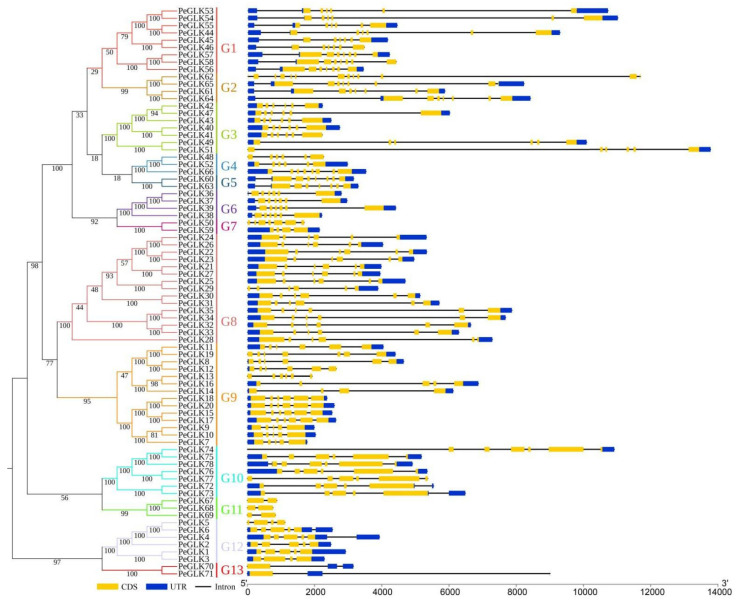
Phylogenetic relationship and gene structures of *PeGLKs*. Left: Neighbor-joining (NJ) phylogenetic tree was constructed by MEGA 7.0 according to PeGLK protein sequences. Proteins in tree were divided into 13 distinct subfamilies, which are distinguished by different colors. Right: Exon/intron structures of *PeGLK* genes were generated by online GSDS. Yellow rectangles, gray lines, and blue rectangles represent exons, introns, and untranslated regions (UTRs), respectively. The numbers at nodes indicate the bootstrap values per 1000 replicates determined by the neighbor-joining method.

**Figure 4 molecules-27-05491-f004:**
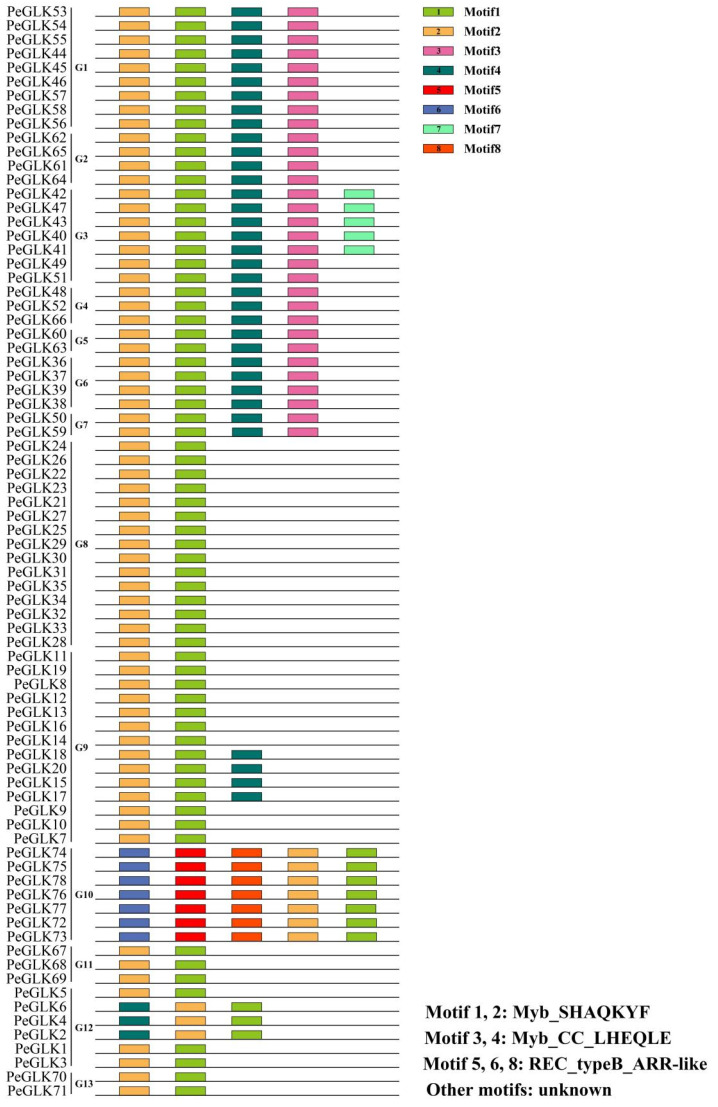
Schematic representation of eight conserved motifs (1–8) in PeGLKs, ordered based on online MEME analysis. Lengths of motifs are displayed proportionally.

**Figure 5 molecules-27-05491-f005:**
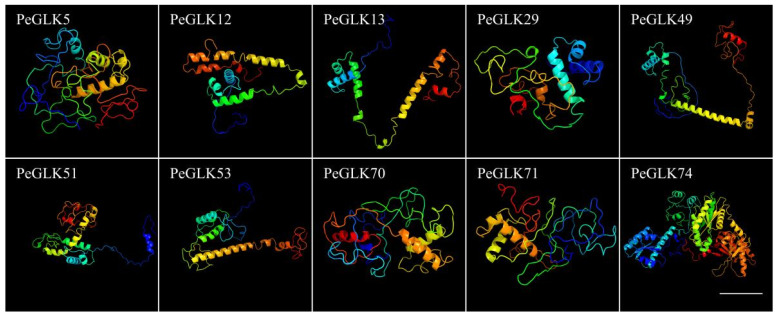
Predicted structures of PeGLK proteins. Structures of 10 PeGLK proteins were predicted with >98% confidence. All colored lines represent α-helix, and the arrow with plane shape is β-sheet, which displayed only in PeGLK74. Bars: 20 nm.

**Figure 6 molecules-27-05491-f006:**
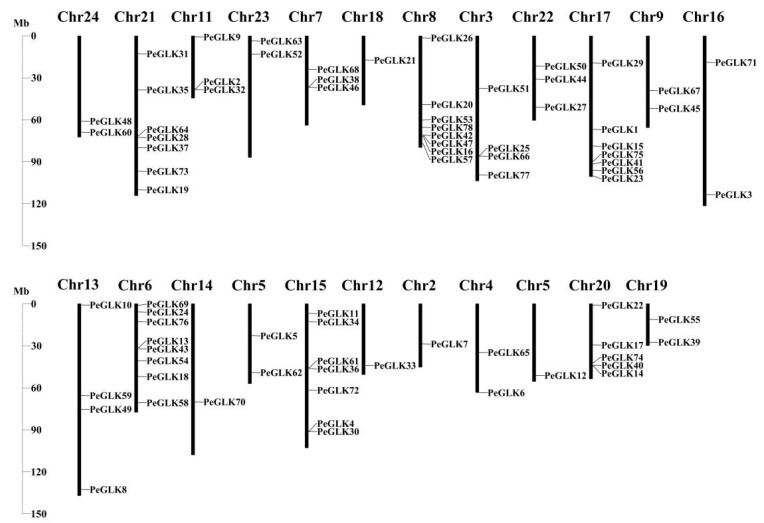
Chromosomal locations of *PeGLK* genes in *Phyllostachys edulis*.

**Figure 7 molecules-27-05491-f007:**
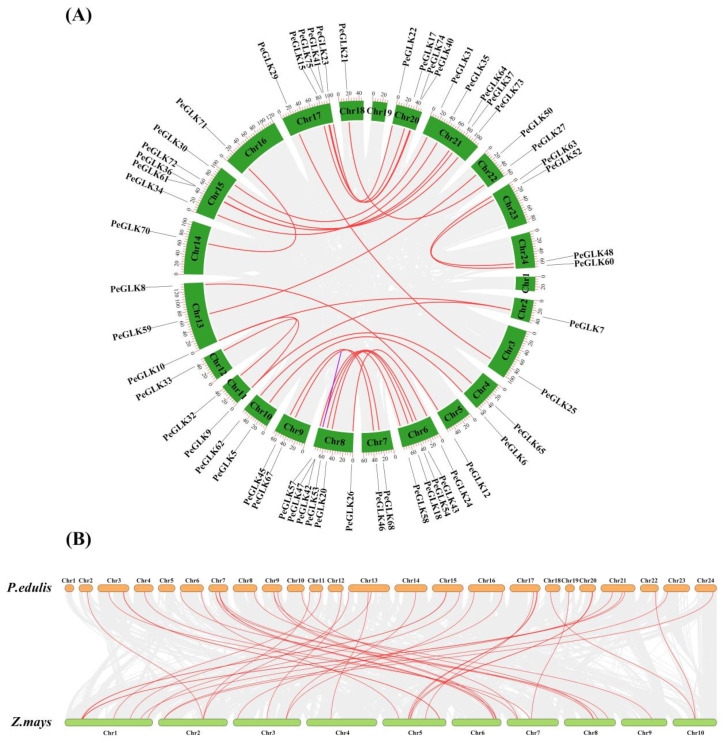
Duplication events of GLK proteins. (**A**) Synteny of moso bamboo *PeGLK* genes. (**B**) Synteny of moso bamboo and maize *GLK* genes. All syntenic genes were located on a map; segmentally duplicated *GLK* gene pairs are linked by red lines and tandemly duplicated genes by purple lines.

**Figure 8 molecules-27-05491-f008:**
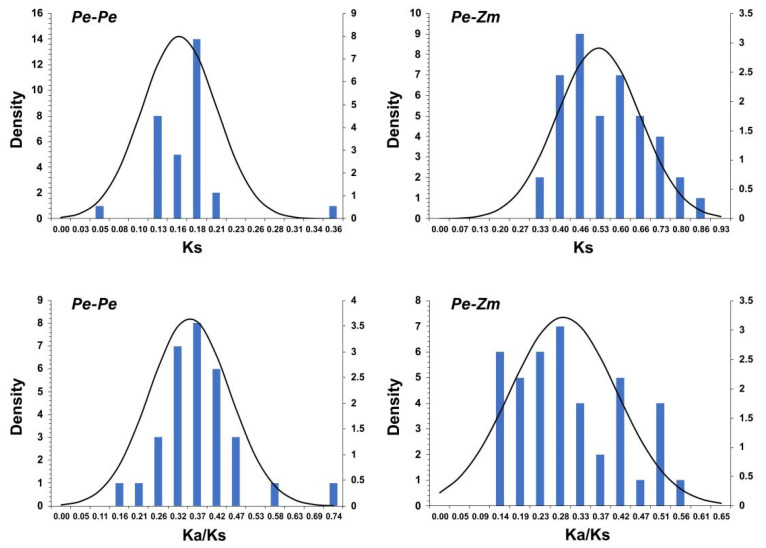
Ks and Ka/Ks value distribution of *PeGLK* genes in paralogous gene pairs (Pe-Pe) of moso bamboo genome and orthologous gene pairs between moso bamboo and maize, viewed through frequency distribution of relative Ks and Ka/Ks modes.

**Figure 9 molecules-27-05491-f009:**
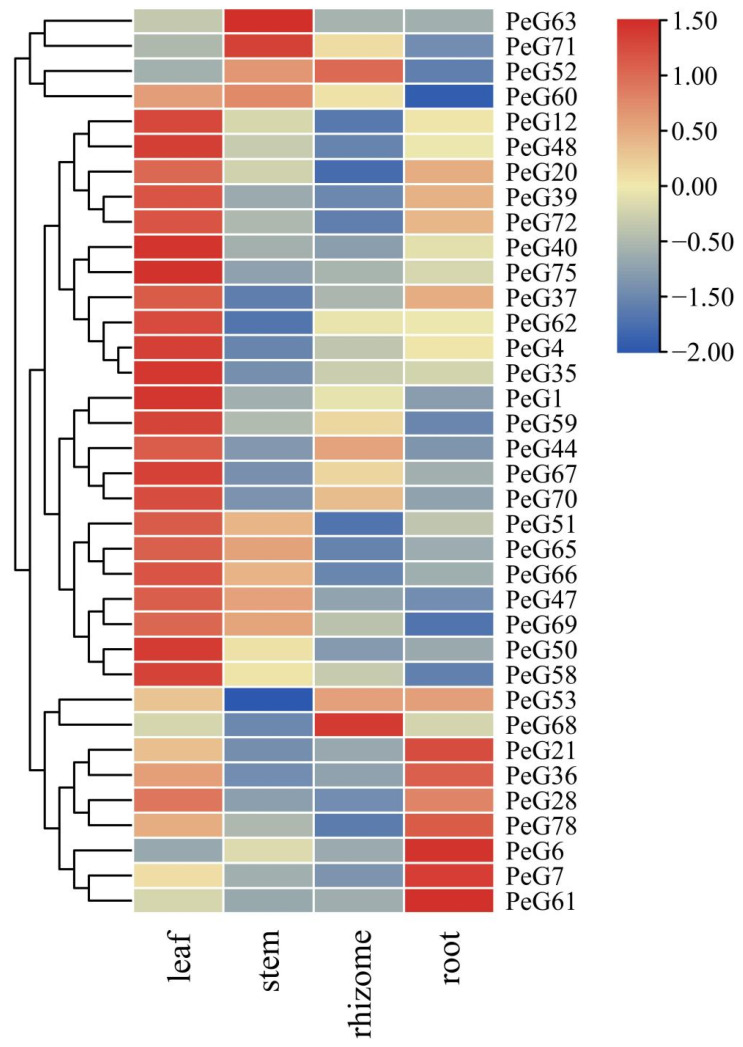
Expression profiles of *PeGLK* genes in different tissues. Samples were from leaf, steam, rhizome, and root. Expression level of each *PeGLK* gene can be estimated based on scale on right. Red, yellow and blue indicate high, medium and low levels of gene expression, respectively.

**Figure 10 molecules-27-05491-f010:**
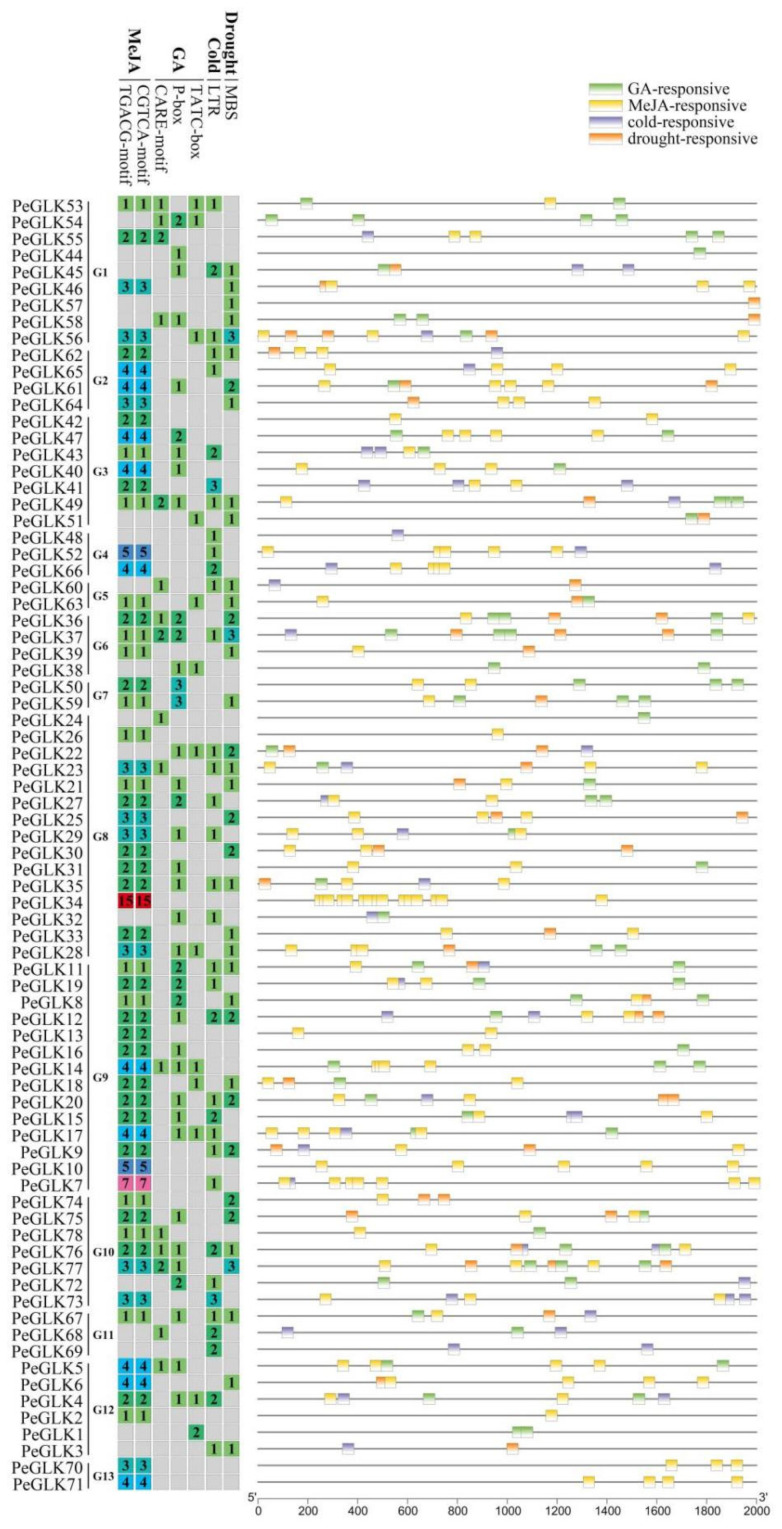
*cis*-Acting elements related to LTRE, DRE, MeJA, and GA in promoter regions of *PeGLKs*. Left: Different colors represent numbers of four *cis*-acting elements of *PeGLKs*. Right: Four CREs were predicted; each is displayed in a different color.

**Figure 11 molecules-27-05491-f011:**
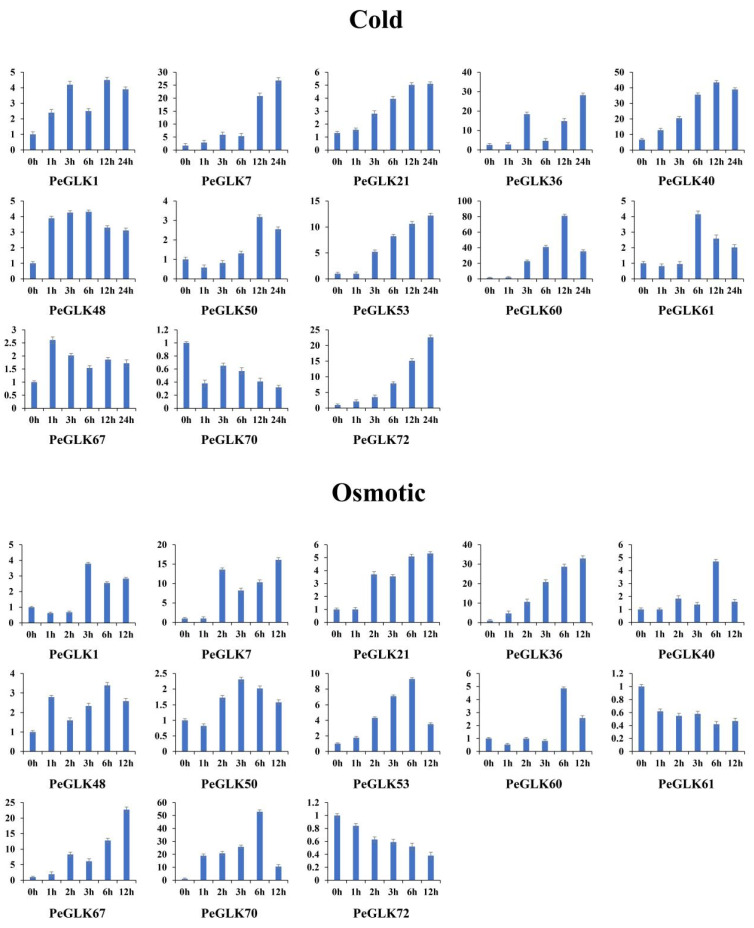
Expression patterns of 13 representative *PeGLK* genes in response to abiotic stress treatments. X-axis and Y-axis indicate time points after cold and osmotic treatments, and relative expression levels are standardized to reference gene *TIP41*.

**Figure 12 molecules-27-05491-f012:**
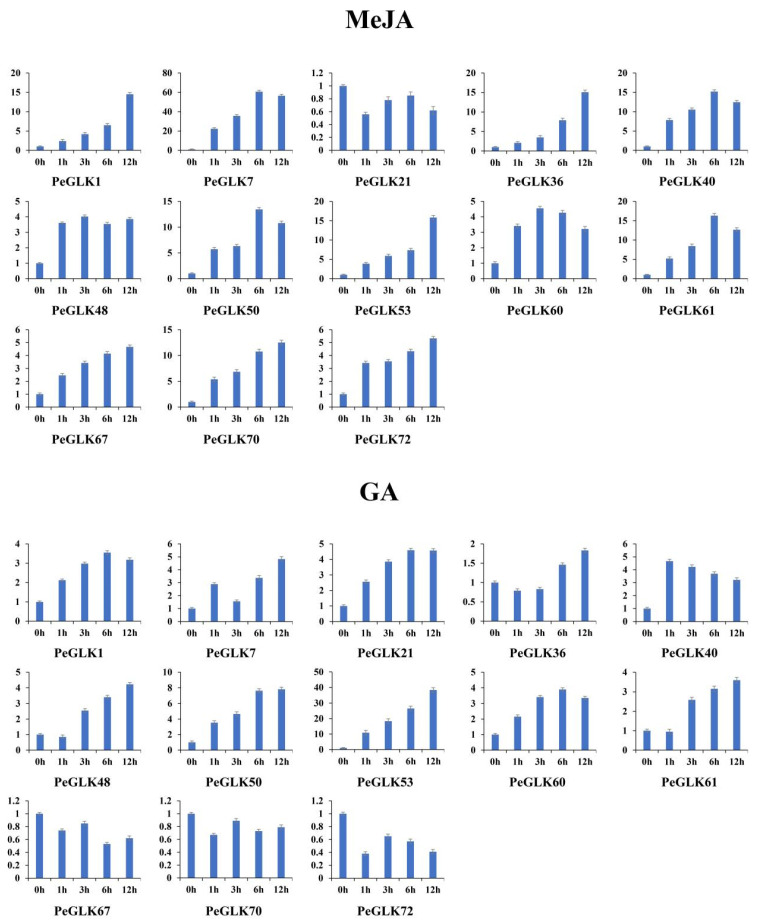
Expression patterns of 13 representative *PeGLK* genes in response to phytohormone treatments. X-axis and Y-axis indicate time points after MeJA and GA treatments, and relative expression levels are standardized to reference gene *TIP41*.

**Table 1 molecules-27-05491-t001:** Detailed information on 78 predicted *PeGLK* genes in *Phyllostachys*
*edulis*. Last column lists gene ID, chromosomal locations, molecular weight (MW), protein isoelectric point (pI), and number of exons in each gene.

Gene Name	Sequences ID	Position	MW (Da)	PI	CDS Length (bp)	Size (aa)	Exons
*PeGLK1*	PH02Gene35128.t1	17:66858322-66861246(+)	42,593.43	8.19	1182	393	5
*PeGLK2*	PH02Gene01193.t1	11:37933106-37935593(+)	38,720.28	7.10	1074	357	5
*PeGLK3*	PH02Gene09030.t2	16:113462311-113464603(−)	49,571.46	6.08	1389	462	4
*PeGLK4*	PH02Gene21002.t1	15:91066421-91070352(+)	38,096.01	6.99	1056	351	5
*PeGLK5*	PH02Gene35342.t1	10:22940475-22941589(+)	27,158.16	8.77	768	255	4
*PeGLK6*	PH02Gene25255.t1	4:63579741-63582272(−)	37,844.34	5.96	1059	352	5
*PeGLK7*	PH02Gene17491.t1	2:28739813-28741586(−)	36,728.56	6.79	1011	336	5
*PeGLK8*	PH02Gene38026.t1	13:132810438-132815080(−)	34,286.6	8.62	927	308	6
*PeGLK9*	PH02Gene18924.t1	11:802821-804806(−)	37,872.77	6.35	1038	345	5
*PeGLK10*	PH02Gene39352.t1	13:873597-875620(−)	38,491.68	6.98	1050	349	5
*PeGLK11*	PH02Gene20517.t1	15:6935099-6939136(−)	37,931.95	8.00	1065	354	6
*PeGLK12*	PH02Gene29515.t1	5:51264001-51266636(−)	33,988.33	8.93	915	304	6
*PeGLK13*	PH02Gene34333.t1	6:31369720-31371637(−)	18,081.66	10.43	486	161	6
*PeGLK14*	PH02Gene30127.t1	20:44761394-44767510(−)	33,657.58	7.08	927	308	4
*PeGLK15*	PH02Gene48395.t1	17:78812464-78814980(−)	56,840.06	6.53	1545	514	6
*PeGLK16*	PH02Gene17440.t1	8:71935937-71942802(+)	28,099.34	8.37	756	251	5
*PeGLK17*	PH02Gene38152.t1	20:29572044-29574673(−)	54,525.46	6.22	1497	498	6
*PeGLK18*	PH02Gene39325.t1	6:52094000-52096363(+)	58,022.99	6.46	1611	536	6
*PeGLK19*	PH02Gene00846.t1	21:110026992-110031388(+)	35059.67	8.95	981	326	7
*PeGLK20*	PH02Gene42607.t1	8:49040798-49043378(−)	57,616.54	5.94	1587	528	6
*PeGLK21*	PH02Gene02579.t2	18:17380009-17383980(+)	36,881.46	7.37	1053	350	6
*PeGLK22*	PH02Gene39580.t2	20:1052803-1058126(−)	43,052.21	9.02	1191	396	6
*PeGLK23*	PH02Gene31338.t2	17:100218030-100222977(+)	41,530.78	9.40	1158	385	6
*PeGLK24*	PH02Gene17108.t1	6:5944617-5949931(+)	34,923.39	9.76	966	321	6
*PeGLK25*	PH02Gene03796.t1	3:85369708-85374396(+)	36,685.24	7.94	1029	342	6
*PeGLK26*	PH02Gene29053.t1	8:1462771-1466790(+)	34,689.98	9.70	951	316	6
*PeGLK27*	PH02Gene49306.t1	22:50874099-50878039(+)	36,221.83	7.77	1029	342	6
*PeGLK28*	PH02Gene34821.t1	21:72609406-72616684(+)	41,826.53	9.19	1209	402	6
*PeGLK29*	PH02Gene32674.t2	17:19419163-19423040(−)	19,477.62	5.95	528	175	6
*PeGLK30*	PH02Gene20991.t1	15:91308193-91313320(+)	38,773.69	6.27	1107	368	6
*PeGLK31*	PH02Gene17776.t1	21:12781867-12787567(−)	38,324.37	6.26	1104	367	6
*PeGLK32*	PH02Gene01179.t1	11:38405451-38412089(+)	42,846.77	7.14	1197	398	6
*PeGLK33*	PH02Gene15844.t1	12:44109361-44115643(+)	42,734.24	7.12	1191	396	6
*PeGLK34*	PH02Gene20627.t1	15:12983299-12990970(−)	42,273.57	9.54	1173	390	6
*PeGLK35*	PH02Gene45340.t1	21:38563232-38571091(−)	41,493.76	9.56	1152	383	6
*PeGLK36*	PH02Gene46403.t1	15:46604970-46607753(−)	39,682.03	8.17	1071	356	6
*PeGLK37*	PH02Gene15930.t1	21:79933476-79936426(+)	39,576.87	8.17	1068	355	6
*PeGLK38*	PH02Gene02408.t1	7:36153335-36155537(−)	46,811.37	6.66	1248	415	6
*PeGLK39*	PH02Gene44186.t1	19:27506582-27510983(−)	39,621.3	8.60	1074	357	6
*PeGLK40*	PH02Gene39987.t1	20:44024199-44026929(+)	37,799.07	8.54	1035	344	6
*PeGLK41*	PH02Gene09101.t1	17:91509553-91511761(+)	38,369.88	9.12	1047	348	6
*PeGLK42*	PH02Gene31703.t1	8:70863226-70865442(+)	30,479.81	8.90	822	273	6
*PeGLK43*	PH02Gene33296.t1	6:32324747-32327222(−)	38,932.24	6.84	1056	351	6
*PeGLK44*	PH02Gene40793.t1	22:30906449-30915724(−)	33,752.45	5.95	924	307	6
*PeGLK45*	PH02Gene05591.t1	9:51866782-51870935(−)	29,354.19	6.67	798	265	6
*PeGLK46*	PH02Gene02376.t1	7:36864743-36868199(−)	29,442.24	7.10	798	265	6
*PeGLK47*	PH02Gene31704.t1	8:70867912-70873914(+)	38,717.97	6.90	1050	349	6
*PeGLK48*	PH02Gene10199.t1	24:60967409-60969661(+)	30,026.13	8.33	822	273	6
*PeGLK49*	PH02Gene09178.t1	13:75454095-75464169(−)	28,140.14	7.78	774	257	6
*PeGLK50*	PH02Gene04584.t1	22:21670037-21671709(−)	37,153.86	6.76	1029	342	7
*PeGLK51*	PH02Gene45566.t1	3:37548923-37570928(−)	28,160.04	8.90	783	260	6
*PeGLK52*	PH02Gene02264.t1	23:13009053-13012018(−)	29,036.04	9.10	795	264	6
*PeGLK53*	PH02Gene13954.t1	8:60262961-60273665(+)	23,380.35	6.60	633	210	6
*PeGLK54*	PH02Gene40923.t1	6:40681881-40692878(−)	36,241.28	6.60	1011	336	6
*PeGLK55*	PH02Gene42198.t1	19:11230444-11234879(−)	31,628.95	6.02	867	288	6
*PeGLK56*	PH02Gene29378.t1	17:96169688-96173122(+)	51,782.72	6.71	1404	467	7
*PeGLK57*	PH02Gene02799.t1	8:77249489-77253699(+)	46,249.31	5.51	1269	422	7
*PeGLK58*	PH02Gene29763.t1	6:70768706-70773109(+)	51,370.07	5.75	1410	469	7
*PeGLK59*	PH02Gene45947.t2	13:65634449-65636584(−)	31,526.73	5.57	858	285	4
*PeGLK60*	PH02Gene00387.t1	24:68886977-68890124(−)	44,369.49	6.73	1209	402	7
*PeGLK61*	PH02Gene19293.t1	15:45733490-45739347(−)	49,605.87	5.85	1356	451	7
*PeGLK62*	PH02Gene11518.t1	10:49175750-49187421(+)	38,777.43	5.61	1068	355	13
*PeGLK63*	PH02Gene20571.t1	23:3653518-3656801(+)	45,464.34	6.74	1230	409	7
*PeGLK64*	PH02Gene12389.t1	21:71256608-71265008(−)	49,101.24	5.98	1350	449	7
*PeGLK65*	PH02Gene36110.t1	4:34815482-34823691(−)	43,616.22	5.02	1215	404	8
*PeGLK66*	PH02Gene03773.t2	3:85969044-85972558(+)	38,144.82	9.58	1017	338	8
*PeGLK67*	PH02Gene21337.t1	9:39183574-39184446(−)	24,896.89	9.51	759	252	2
*PeGLK68*	PH02Gene22746.t1	7:23963609-23964367(−)	22,731.56	9.04	672	223	2
*PeGLK69*	PH02Gene10071.t1	6:1102829-1103662(+)	20,032.6	6.06	555	184	2
*PeGLK70*	PH02Gene11430.t1	14:70153233-70156386(+)	23,995.12	6.61	675	224	1
*PeGLK71*	PH02Gene20674.t1	16:18948494-18957497(−)	24,222.27	6.25	684	227	1
*PeGLK72*	PH02Gene11768.t1	15:61838091-61843627(−)	74,154.2	5.79	2070	689	5
*PeGLK73*	PH02Gene26606.t1	21:96791235-96797716(−)	74,480.85	5.86	2070	689	5
*PeGLK74*	PH02Gene30734.t1	20:42027130-42038041(+)	67,709.51	6.06	1860	619	6
*PeGLK75*	PH02Gene04797.t1	17:89520790-89525967(+)	68,432.23	6.11	1881	626	6
*PeGLK76*	PH02Gene11687.t1	6:12955225-12960572(+)	74,366.97	5.98	2031	676	6
*PeGLK77*	PH02Gene21205.t1	3:99493453-99498815(−)	74,410.85	6.04	2058	685	6
*PeGLK78*	PH02Gene33414.t1	8:65390201-65395109(+)	74,995.96	5.59	2079	692	6

**Table 2 molecules-27-05491-t002:** Ka/Ks and divergence analysis of paralogous *PeGLK* genes in moso bamboo.

Paralogous	Ka	Ks	Ka/Ks	Selection Pressure	Duplicate Type
*PeGLK5/6*	0.04	0.12	0.31	Purifying selection	Segmental duplication
*PeGLK7/10*	0.07	0.19	0.37	Purifying selection	Segmental duplication
*PeGLK7/9*	0.08	0.18	0.46	Purifying selection	Segmental duplication
*PeGLK9/10*	0.07	0.17	0.39	Purifying selection	Segmental duplication
*PeGLK8/12*	0.05	0.12	0.45	Purifying selection	Segmental duplication
*PeGLK15/17*	0.09	0.12	0.72	Purifying selection	Segmental duplication
*PeGLK18/20*	0.08	0.20	0.39	Purifying selection	Segmental duplication
*PeGLK21/27*	0.06	0.18	0.30	Purifying selection	Segmental duplication
*PeGLK22/23*	0.04	0.12	0.33	Purifying selection	Segmental duplication
*PeGLK24/26*	0.04	0.18	0.25	Purifying selection	Segmental duplication
*PeGLK25/29*	0.03	0.18	0.14	Purifying selection	Segmental duplication
*PeGLK30/31*	0.05	0.14	0.35	Purifying selection	Segmental duplication
*PeGLK32/33*	0.06	0.17	0.37	Purifying selection	Segmental duplication
*PeGLK34/35*	0.04	0.12	0.36	Purifying selection	Segmental duplication
*PeGLK36/37*	0.03	0.18	0.19	Purifying selection	Segmental duplication
*PeGLK40/41*	0.03	0.13	0.24	Purifying selection	Segmental duplication
*PeGLK42/43*	0.05	0.18	0.27	Purifying selection	Segmental duplication
*PeGLK43/47*	0.05	0.17	0.28	Purifying selection	Segmental duplication
*PeGLK45/46*	0.04	0.12	0.31	Purifying selection	Segmental duplication
*PeGLK48/52*	0.05	0.17	0.30	Purifying selection	Segmental duplication
*PeGLK50/59*	0.06	0.17	0.34	Purifying selection	Segmental duplication
*PeGLK53/54*	0.05	0.14	0.33	Purifying selection	Segmental duplication
*PeGLK57/58*	0.06	0.17	0.34	Purifying selection	Segmental duplication
*PeGLK60/63*	0.06	0.18	0.34	Purifying selection	Segmental duplication
*PeGLK61/64*	0.05	0.13	0.40	Purifying selection	Segmental duplication
*PeGLK62/65*	0.20	0.36	0.57	Purifying selection	Segmental duplication
*PeGLK67/68*	0.03	0.12	0.24	Purifying selection	Segmental duplication
*PeGLK70/72*	0.08	0.18	0.46	Purifying selection	Segmental duplication
*PeGLK71/73*	0.04	0.12	0.33	Purifying selection	Segmental duplication
*PeGLK74/75*	0.04	0.13	0.27	Purifying selection	Segmental duplication
*PeGLK42/47*	0.02	0.05	0.40	Purifying selection	Tandem duplication

**Table 3 molecules-27-05491-t003:** Ka/Ks and divergence analysis of orthologous *GLK* genes in moso bamboo and maize.

Paralogous	Ka	Ks	Ka/Ks	Selection Pressure
*ZmGLK27/PeGLK66*	0.09	0.79	0.11	Purifying selection
*ZmGLK48/PeGLK38*	0.10	0.56	0.17	Purifying selection
*ZmGLK9/PeGLK2*	0.14	0.59	0.23	Purifying selection
*ZmGLK2/PeGLK61*	0.16	0.41	0.38	Purifying selection
*ZmGLK2/PeGLK64*	0.17	0.42	0.40	Purifying selection
*ZmGLK1/PeGLK36*	0.08	0.65	0.13	Purifying selection
*ZmGLK1/PeGLK37*	0.08	0.71	0.12	Purifying selection
*ZmGLK6/PeGLK35*	0.25	0.69	0.36	Purifying selection
*ZmGLK28/PeGLK41*	0.07	0.38	0.18	Purifying selection
*ZmGLK28/PeGLK40*	0.07	0.38	0.18	Purifying selection
*ZmGLK34/PeGLK75*	0.14	0.59	0.24	Purifying selection
*ZmGLK34/PeGLK74*	0.15	0.64	0.23	Purifying selection
*ZmGLK32/PeGLK25*	0.14	0.57	0.25	Purifying selection
*ZmGLK43/PeGLK51*	0.08	0.53	0.14	Purifying selection
*ZmGLK49/PeGLK69*	0.13	0.49	0.27	Purifying selection
*ZmGLK20/PeGLK70*	0.27	0.50	0.53	Purifying selection
*ZmGLK20/PeGLK71*	0.23	0.45	0.51	Purifying selection
*ZmGLK51/PeGLK42*	0.14	0.42	0.32	Purifying selection
*ZmGLK29/PeGLK58*	0.11	0.55	0.19	Purifying selection
*ZmGLK3/PeGLK49*	0.11	0.41	0.26	Purifying selection
*ZmGLK39/PeGLK46*	0.06	0.50	0.13	Purifying selection
*ZmGLK39/PeGLK45*	0.06	0.50	0.12	Purifying selection
*ZmGLK59/PeGLK21*	0.12	0.55	0.21	Purifying selection
*ZmGLK59/PeGLK27*	0.12	0.42	0.28	Purifying selection
*ZmGLK10/PeGLK4*	0.17	0.60	0.28	Purifying selection
*ZmGLK13/PeGLK52*	0.16	0.41	0.40	Purifying selection
*ZmGLK13/PeGLK48*	0.16	0.46	0.36	Purifying selection
*ZmGLK44/PeGLK62*	0.20	0.46	0.43	Purifying selection
*ZmGLK44/PeGLK65*	0.12	0.38	0.32	Purifying selection
*ZmGLK26/PeGLK59*	0.20	0.39	0.51	Purifying selection
*ZmGLK45/PeGLK55*	0.09	0.56	0.17	Purifying selection
*ZmGLK19/PeGLK3*	0.13	0.35	0.39	Purifying selection
*ZmGLK18/PeGLK33*	0.32	0.84	0.39	Purifying selection
*ZmGLK41/PeGLK68*	0.07	0.31	0.24	Purifying selection
*ZmGLK47/PeGLK67*	0.07	0.37	0.20	Purifying selection
*ZmGLK50/PeGLK68*	0.10	0.33	0.31	Purifying selection
*ZmGLK50/PeGLK67*	0.08	0.38	0.20	Purifying selection
*ZmGLK31/PeGLK15*	0.25	0.79	0.32	Purifying selection
*ZmGLK37/PeGLK38*	0.10	0.64	0.15	Purifying selection
*ZmGLK17/PeGLK9*	0.31	0.64	0.48	Purifying selection
*ZmGLK17/PeGLK10*	0.34	0.70	0.49	Purifying selection
*ZmGLK17/PeGLK7*	0.33	0.70	0.47	Purifying selection

## Data Availability

Not applicable.

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
