# Peer review of "Genome-Wide Identification and Characterization of *G2-Like* Transcription Factor Genes in Moso Bamboo (*Phyllostachys edulis*)"

_molecules, 2022, doi:10.3390/molecules27175491_

Round 1
Reviewer 1 Report
Dear Authors,
Reviewer comments molecules-1865132
The manuscript entitled „Genome-wide identification and characterization of G2-like transcription factor genes in moso bamboo“ represents a useful bioinformatics and expression analysis study on moso bamboo (Phyllostachys edulis) transcription factors including their structure, phylogenetic and expression analyses in response to abiotic stress and phytohormone treatments.
I have no major comment onthe present manuscript. However, I have a few important comments o the present version of the manuscript:
1/ In Materials and methods, line 95, the authors should clearly declare that they searched the moso bamboo GLK proteins in NCBI protein database as it is evident from the web address related to BLASTP search and they have to add the date of access to the information on NCBI database since the protein sequences in NCBI database are updated with time so the information on the date of access is important.
2/ The plant scientific name, i.e., Phyllostachys edulis, not only in Abstract, but also in Materials and methods, Plant materiál section, and in the manuscript title, following „moso bamboo“.
3/ Terminology:
The authors should use the term „osmotic stress“ instead of „drought stress“ for PEG-6000 treatment since „drought“ means a water deficit in the soil while plant treatment with 20% PEG-6000 causes an osmotic stress due to decreased osmotic potential of the growth medium.
4/ Results:
In Figure 2 and Figure 3 legends aimed at the description the phylogenetic tree, a short description of the numbers at nodes has to be added to the legends, i.e., „The numbers at nodes indicate the bootstrap values per 1000 replicates determined by the neighbour-joining method.“
In Figure 5 providing models of 10 PeGLK proteins, appropriate scale bars have to be added.
5/ In Materials and methods, part 2.1., line 79: I do not understand the statement „100-day-old moso bamboo seeds“ ?? Does it mean 100 days after the harvest?? Maybe that the statement „100-day-old“ should be replaced to the plants subjected to the abiotic stress and phytohormone treatments, i.e., that „100-day-old plants were subjected to cold, osmotic stress, ….“
Regarding the 20% PEG-6000 treatment, were the plants treated with pure water with 20% PEG-6000 or some were some mineral nutrients added to the growth medium? Maybe that it was only water since the treatment was short, just only 12 h, but this information has to be given in Materials and methods.
6/ Formal comments on the text:
Introduction, line 32: Add the word „is“ in the statement „Chloroplasts contain the green pigment chlorophyll which can carry out photosynthesis and is thus essential in plant life processes.“
Materials and methods, lines 83, 89: Replace the verb „were irrigated“ with „were treated“ since „irrigation“ means only watering, i.e., treatment with water while other treatments are mentioned in the text.
Materials and methods, line 131: Modify the word form „to product“ to „to produce“, i.e., „MCScanX software was used to produce collinearity blocks…“
Materials and methods, line 143: Use the verb „to predict“ instead of „ro prognosticate“, i.e., „…was used to predict the putative cis-elements…“
Results, line 194: Modify the word form „classed“ to „classified“ in the statement „GLK family members were classified into 16 groups…“
Results, line 204 and further in the text: Modify the word form „in accord“ to „in accordance“ in the statement „PeGLK proteins were also classified into 13 distinct subfamilies, in accordance with the phylogenetic tree…“
Results, line 256: Add the specification „PeGLK“ inthe statement „Chr13 was the longest chromosome with only four PeGLK genes.“
Discussion, Line 412: Correct the term „syntheny analysis“ (not „synthesis analysis“).
Discussion, Line 434: Modify the word form „highly“ to „high“ in the statement „… that great majority of the PeGLK genes exhibited high expression levels in the leaf.“
Discussion, line 449: Modify the statement as follows: „…in response to abiotic stresses and phytohormone treatments“, NOT „in response to abiotic and phytohormone stresses“!! – the words „phytohormone stresses“ is unacceptable in the scientific text!!
Conclusions, line 471: Add the word „stresses“ following the word „abiotic“ in the statement „…under different abiotic stresses and phytohormone treatments“.
Line 498: In Abbreviations list, the abbreviations have to be ordered alphabetically following standard English alphabet.
Final recommendation: Accept after a minor revision.
Author Response
Question 1: In Materials and methods, line 95, the authors should clearly declare that they searched the moso bamboo GLK proteins in NCBI protein database as it is evident from the web address related to BLASTP search and they have to add the date of access to the information on NCBI database since the protein sequences in NCBI database are updated with time so the information on the date of access is important..
Answer: Thank you for your affirmation of our work. We sincerely accept this comment and further add the date of access to the information on NCBI database. Changes made are now visible in lines 100.
Question 2: The plant scientific name, i.e., Phyllostachys edulis, not only in Abstract, but also in Materials and methods, Plant materiál section, and in the manuscript title, following „moso bamboo“.
Answer: Thanks for this helpful suggestion and we sincerely accept this comment. We have added Phyllostachys edulis after moso bamboo associated along with the manuscript.
Question 3: Terminology: The authors should use the term „osmotic stress“ instead of „drought stress“ for PEG-6000 treatment since „drought“ means a water deficit in the soil while plant treatment with 20% PEG-6000 causes an osmotic stress due to decreased osmotic potential of the growth medium.
Answer: Thanks for this helpful suggestion and we sincerely accept this comment. Changes made are now visible in lines 25, 87-90 and 361-398.
Question 4: Results: In Figure 2 and Figure 3 legends aimed at the description the phylogenetic tree, a short description of the numbers at nodes has to be added to the legends, i.e., „The numbers at nodes indicate the bootstrap values per 1000 replicates determined by the neighbour-joining method.“
In Figure 5 providing models of 10 PeGLK proteins, appropriate scale bars have to be added.
Answer: Thanks for this helpful suggestion and we sincerely accept this comment. Changes made are now visible in lines 213-214, 235-236, and have already added the appropriate scale bars in Figure 5.
Question 5: In Materials and methods, part 2.1., line 79: I do not understand the statement „100-day-old moso bamboo seeds“ ?? Does it mean 100 day after the harvest?? Maybe that the statement „100-day-old“ should be replaced to the plants subjected to the abiotic stress and phytohormone treatments, i.e., that „100-day-old plants were subjected to cold, osmotic stress, ….“
Regarding the 20% PEG-6000 treatment, were the plants treated with pure water with 20% PEG-6000 or some were some mineral nutrients added to the growth medium? Maybe that it was only water since the treatment was short, just only 12 h, but this information has to be given in Materials and methods.
Answer: Thanks for this helpful suggestion and we really appreciate this suggestion, 100-day-old moso bamboo seeds means moso bamboo seeds grown for 100 days. In addition, 20% PEG-6000 treatment means that the plants are treated with pure water containing 20% PEG-6000. Changes made are now visible in lines 86-90.
Question 6: 6/ Formal comments on the text:
Introduction, line 32: Add the word „is“ in the statement „Chloroplasts contain the green pigment chlorophyll which can carry out photosynthesis and is thus essential in plant life processes.“
Materials and methods, lines 83, 89: Replace the verb „were irrigated“ with „were treated“ since „irrigation“ means only watering, i.e., treatment with water while other treatments are mentioned in the text.
Materials and methods, line 131: Modify the word form „to product“ to „to produce“, i.e., „MCScanX software was used to produce collinearity blocks…“
Materials and methods, line 143: Use the verb „to predict“ instead of „ro prognosticate“, i.e., „…was used to predict the putative cis-elements…“
Results, line 194: Modify the word form „classed“ to „classified“ in the statement „GLK family members were classified into 16 groups…“
Results, line 204 and further in the text: Modify the word form „in accord“ to „in accordance“ in the statement „PeGLK proteins were also classified into 13 distinct subfamilies, in accordance with the phylogenetic tree…“
Results, line 256: Add the specification „PeGLK“ in the statement „Chr13 was the longest chromosome with only four PeGLK genes.“
Discussion, Line 412: Correct the term „syntheny analysis“ (not „synthesis analysis“).
Discussion, Line 434: Modify the word form „highly“ to „high“ in the statement „… that great majority of the PeGLK genes exhibited high expression levels in the leaf.“
Discussion, line 449: Modify the statement as follows: „…in response to abiotic stresses and phytohormone treatments“, NOT „in response to abiotic and phytohormone stresses“!! – the words „phytohormone stresses“ is unacceptable in the scientific text!!
Conclusions, line 471: Add the word „stresses“ following the word „abiotic“ in the statement „…under different abiotic stresses and phytohormone treatments“.
Line 498: In Abbreviations list, the abbreviations have to be ordered alphabetically following standard English alphabet.
Answer: Thanks for this helpful suggestion and we sincerely accept this comment. Changes made are now visible in lines 33, 94, 137, 149, 206, 217, 271, 437, 477, 505, and 532-535.
Reviewer 2 Report
- The manuscript titled "Genome-Wide Identification and Characterization of G2-Like Transcription Factor Genes in Moso Bamboo" addressed the genome-wide identification, phylogenetic relationships, conserved motifs, promoter cis-elements, MCScanX, divergence times, and expression profile analysis of PeGLK genes in moso bamboo (Phyllostachys edulis). Overall, 78 putative PeGLKs (PeGLK1–PeGLK78) were identified and divided into 13 distinct subfamilies. Each subfamily contains members displaying similar gene structure and motif composition. By synteny analysis, 42 orthologous pairs and highly conserved microsynteny between regions of GLK genes across moso bamboo and maize were found. Furthermore, an analysis of the divergence times indicated that PeGLK genes had a duplication event around 15 million years ago (MYA) and a divergence happened around 38 MYA between PeGLK and ZmGLK. Tissue-specific expression analysis showed that PeGLK genes presented distinct expression profiles in various tissues, and many members were highly expressed in leaves. Additionally, several PeGLKs were significantly up-regulated under cold stress, drought stress, and MeJA and GA treatment, implying that they have a likelihood of affecting abiotic stress and phytohormone responses in plants.
- The experimental work and the results are interesting and could provide a comprehensive understanding of the moso bamboo GLK gene family, as well as elucidating the potential functional characterization of PeGLK genes.
Only minor revisions are requested as follows;
- The methods needs more details such as expression analyses and measured parameters.
-The resolution of Figure 3 and Figure 4 are low. It would be better if the authors could provide high quality figures to be more clear and visible.
- The discussion should include one more paragraph discussing the overall mechanisms assayed in the present study in relation to the previous findings and literature.
- The conclusions section should be re-written to highlight the most significant findings and future recommended research work to further enhance the work.
Author Response
Question 1: The methods needs more details such as expression analyses and measured parameters.
Answer: Thanks for this helpful suggestion and we sincerely accept this comment and have already added more details such as expression analyses and measured parameters in lines 114, 156-157, and 168-170.
Question 2: The resolution of Figure 3 and Figure 4 are low. It would be better if the authors could provide high quality figures to be more clear and visible.
Answer: Thanks for this helpful suggestion and we sincerely accept this comment and have already changed high quality figures to be more clear and visible.
Question 3: The discussion should include one more paragraph discussing the overall mechanisms assayed in the present study in relation to the previous findings and literature.
Answer: Thanks for this helpful suggestion and we sincerely accept this comment and have already added a paragraph to discuss the overall mechanisms assayed in the present study in relation to the previous findings and literature. Changes made are now visible in lines 401-409.
Question 4: The conclusions section should be re-written to highlight the most significant findings and future recommended research work to further enhance the work.
Answer: Thanks for this helpful suggestion and we sincerely accept this comment and have already re-written the conclusions section to highlight the most significant findings and future recommended research work. Changes made are now visible in lines 494-507.